

# Evaluating soil erosion and sediment deposition rates by the [137]Cs fingerprinting technique at karst gabin basin in Yunnan Province, southwest China

Yanqing Li[1,2,3], Zhongcheng Jiang [2,3], Zhihua Chen[1], Yang Yu [4]*, Funing Lan[2,3], Xiangfei Yue[2,3], Peng
Liu[2,3] Jesús Rodrigo-Comino[5,6]

[1] China University of Geosciences, Wuhan 430074, China

[2] Institute of Karst Geology, Chinese Academy of Geological Sciences, Guilin 541004, China

[3] Key Laboratory of Karst Ecosystem and Treatment of Rocky Desertification, Guilin 541004, China

[4] Department of Sediment Research, China Institute of Water Resources and Hydropower Research, Beijing,100038, China

[5] Department of Physical Geography, University of Trier, 54296 Trier, Germany

[6] Soil Erosion and Degradation Research Group, Department of Geography, Valencia University, Blasco Ibàñez, 28, 46010
Valencia, Spain

*Correspondence to*: Yang Yu (theodoreyy@gmail.com)

**Abstract.** Soil erosion is a global environmental problem that can lead to the loss of nutrients in topsoil layers, particularly in

fragile karst environments where the low contents of organic carbon and steep slopes used to be key pedological and

geomorphological factors. Researching the erosion and deposition rates in small watersheds is important for designing efficient

soil and water conservation measures. In this research, the Dapotou closed catchment, a representative depression in karst

gabin basin, located in the Yunnan province, Southwest China, was selected to assess the variation of soil erosion and soil

sediment mobilisation at different hillslope positions using the [137]Cs tracing technique. The results showed that the soil erosion

rates in the shoulders, backslopes and footslopes were 0.87, 0.35 and 0.49 cm $a^{-1}$, respectively, meanwhile the soil sediment

rate in depression bottom was 2.65 cm $a^{-1}$. The average annual soil erosion modulus of the complete hillslope was 632 t $km^{-2}a^{-1}$, which confirmed in the serious gradation according to karst soil erosion standards. The soil deposition modulus reached

up to 3180 t $km^{-2}a^{-1}$. The sediment delivery ratio summarized 0.82 in the whole catchment according to the square of hillslope

and depression bottom. To identify which factor could play the most important role, a Principal Component Analysis was

conducted. The results showed [137]Cs concentration of different soil depth at different hillslope positions were significant

correlated with soil organic matter (SOM) and total nitrogen (TN) ($P<0.05$). As the typical karst geomorphological types, these

findings are expected to provide data support for the whole watershed soil erosion management and ecological restoration in

this fragile karst ecosystem.

## 1 Introduction

Soil erosion has been identified as a global geo-environmental hazard (Prise et al., 2009; Panagos et al., 2014). Due to the

nutrient-poor characteristics of carbonate-rock parent materials, the effects of soil erosion have been greater in the karst areas,

which occupy approximately 12% of global continental terrains (Febles-González et al., 2012). Considering the 22 million

$km^2$ of global karst areas, a total of 15.6% is located in China, amounting approximately 3.44 million $km^2$ (36% of China).

Karst areas in China are mainly concentrated in eight southwest provinces, including Yunnan, Guizhou, Guangdong,





Chongqing, Hunan, Hubei, Sichuan and Guangxi (Jiang et al., 2014).

In karst areas, gabin basins and mountains usually coexist because of specific geological processes such as subsidence and dissolution of fault blocks induced by Cenozoic tectonic uplift among others (Wang et al., 2017). These differences in geomorphological characteristics, in addition to severe anthropogenic disturbance (e.g., agriculture), often produce a series of

environmental problems (e.g., runoff, erosion). In the gabin basins of China, studies on soil erosion and sediment yield considering small catchments (10–10,000 ha) remains absent, although they are particularly interesting for the understanding of the linkages between soil erosion on hillslopes and sediment transport in large watershed and their potential impacts on the ecological services and human activities. (Schuller et al., 2004).

Since the last decades, many scholars are trying to use $^{137}$Cs to evaluate long-term soil erosion or soil sediment mobilization

in karst areas, whose studies focused on karst peak cluster depression and karst plateau (Bai et al., 2010; Feng et al., 2016; Luo et al., 2018). The $^{137}$Cs is an artificial radionuclide with a half-life of 30.17 years, which is released into the atmosphere as a result of thermo-nuclear weapon testing between 1950s and 1970s (IAEA2014). Traces of $^{137}$Cs enter the earth's soils through dry and wet deposition, with the maximum deposition rate occurring in 1963 (Zhang et al., 2008). $^{137}$Cs has been a cost- and time-effective tool to evaluate soil redistribution due to erosion and can complement the information provided by conventional

erosion measurements (Lizaga et al., 2019), for example in situ experiments such as rainfall, runoff or wind simulations (Marzen et al., 2019; Rodrigo-Comino et al., 2017 and soil erosion plots (Cerdà et al., 2018; Kinnell et al., 2016)$^{137}$Cs can provide retrospective estimates of long-term soil erosion and deposition rates without disturbing the soil environment by installing measuring equipment (Porto and Walling, 2012). In addition, it can also be used to obtain detailed analyses of sediment migration on hillslopes (Evans et al., 2019; Zebari et al., 2019).

In the small catchment of eastern Yunnan, where it has been adversely impacted by soil erosion, there is an urgent need to assess the sediment delivery ratio based on soil erosion and sediment rates in order to inform the policymakers and stakeholders about the potential land degradation processes and possible control measures that should be included. Therefore,using $^{137}$Cs, we aimed to quantify the hillslope soil erosion, soil sediment mobilisation and sediment delivery in this closed depression catchment of the gabin basin. Generally, the concentrations of deposited $^{137}$Cs are usually uniform in a small catchment, which

has similar latitude and rainfall characteristics (Song et al., 2018). Our study pretend to detect the variations of $^{137}$Cs and soil properties under different positions on karst hillslopes, which are representative geological structure in southwest China. In addition, we attempt to quantify the impact of soil depths and positions on soil erosion rates and to analyze the relationship between soil property variations and erosion rates using multivariate statistical techniques. We hypothesize that the results obtained from this research will support the necessary information required to protect the whole watershed of Yunnan karst

gabin basin.



## 2 Materials and methods

### 2.1 Study area

The Dapotou depression is located in the Yangjie Town, Kaiyuan County, situated in the Yunnan Province from China (103°

17′ 25.63"-103°18′ 3.40" E, 23° 36′ 48.04"-23° 37′ 28.10"N) (Fig. 1). This territory is a closed catchment with a drainage area

of about 1.97 km$^2$. Its elevations range from 1267 to 1413 m a.s.l. The underlying bedrock of the depression is a Triassic

carbonate rock, which consists of Gejiu Group ($T_2g^3$) and Falang Group ($T_2f^1$) limestone. Most soils in this watershed have a

soil texture of clay limestone materials.


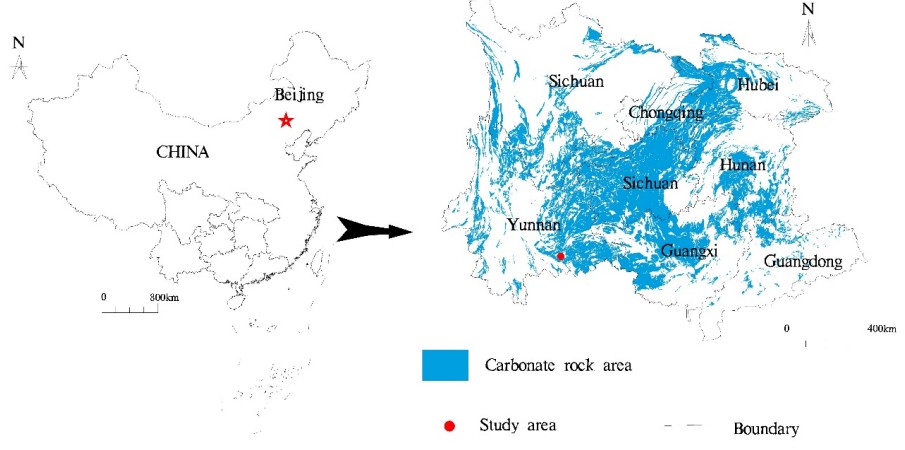

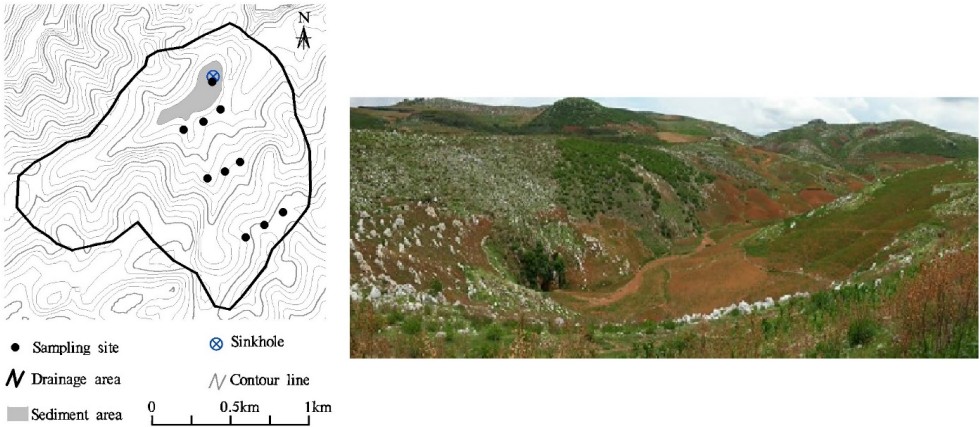

**Figure 1: Localisation of the study area, sampling points and panoramic image of one selected plot.**

This territory experiences a subtropical monsoonal climate with two main seasons: a rainy season from June to September and

a dry season from October to May (Jiang, 2012). The mean annual precipitation is 904 mm with an unimodal rainfall regime.

The rainy season precipitation accounts for 80 % of the annual precipitation. The average annual temperature is 18.3°C (the coordinates of the climate station is 103°19′12.76″E , 23°37′17.25″N).

The study area has a depression bottom with a length of about 1000 m and an average width of about 60 m. The hillslopes and
depression bottom itself are once-a-year cultivated with maize. There is a sinkhole in northeast of the bottom of depression (Fig. 1). Some steep slope areas are dominated by calcicole shrub and drought tolerant herbs. When experiencing a heavy rain, the bottom of the depression is submerged because the sinkhole is not able to canalize the total amount of runoff. Some part of the surface water infiltrates to the soil, and some other flow into the subterranean stream through the sinkhole. It uses to be at least once or twice flood events every rainy season.

**2.2 Sampling tools and sampling design**

In this study, samples in nine cultivated hillslopes at three different hillslope positions (shoulder, backslope and footslope) and one at the depression bottom were collected in July 2017. Soil samples were taken considering a depth of 40 cm on each hillslope position (although in one site of backslope was sampled only to 30 cm deep because of the soil depth) and 240 cm at the depression bottom to ensure that the full radionuclide soil content was taken into account. To establish the vertical
distribution of $^{137}$Cs, the samples were collected by using a scraper with 5 cm increments. A total of 70 soil samples from the hillslope and 48 soil samples from the depression. The bulk density of soil samples was measured by using a cutting ring. Geographical coordinates and elevation of each sampling point were recorded by using a GPS device. After that, soil samples were transported to the laboratory and were sieved through 2 mm sieve to remove plant roots before air drying and performing the radionuclide and physicochemical property analyses.

We also select a control plot as a reference of the inventory site. Ideally, the land for the reference inventory should be flat and undisturbed. However, it is difficult to find any completely flat land which has not been cultivated since the mid-1950s. Therefore, local reference samples of $^{137}$Cs were collected in a relatively flat shrub-grassland site, located several kilometers from the study area(103°26′48.37″E, 23°29′20.15″N). The reference site is well vegetated and protected (undisturbed by the local residents). Five soil samples were then collected in this reference site and the average soil thickness of reference profile
was estimated close to 20 cm.

**2.3 Laboratory analysis**

The $^{137}$Cs analyses were performed in the Institute of Mountain Hazard and Environment, Chinese Academy of Sciences. The $^{137}$Cs content of the <2 mm fraction of each sample was measured by γ spectrometry using a hyper pure coaxial germanium detector and multichannel analyzer system. The samples have a weight of 300 g. $^{137}$Cs was detected at 662 keV and counting
times were more than 50 000 s, providing results with an analytical precision of approximately ±5% at the 95% level of confidence.

Soil physicochemical properties were measured in Northwest Institute of Eco-Environment and Resources, Chinese Academy



of Sciences. Soil organic matter (SOM) was determined by dry combustion at 500 °C (Davis, 1974). Total nitrogen (TN) and

total phosphorus (TP) concentration were measured by using persulfate digestion method. Total potassium (TK) was measured

by using flame photometry method. Soil pH was analyzed by using Mettler Toledo Seven Excellence pH meter.

### 2.4 Conversion models

The $^{137}$Cs inventory of soil profile in different hillslope position was calculated by using the following equation (Zhang et al.,

2009b):

$$CPI = \sum_{i=1}^{n} Ci \times Bdi \times Di \times 10 \tag{1}$$

Where CPI is the $^{137}$Cs inventory (Bq m$^{-2}$), represents the total amount of $^{137}$Cs in sample; $i$ is the sampling layer sequence; n

is the quantity of sampling layer; $C_i$ is $^{137}$Cs concentration (Bq kg$^{-1}$); $Bd_i$ is the bulk density of the $i$ layer (g cm$^{-3}$) and $D_i$ is

the depth of $i$ layer (cm).

Reference sample was considered using a bulk sample and the $^{137}$Cs inventory was calculated following this equation (Zhang

et al., 2009b):

$$CPI=C_i \times W/S \tag{2}$$

Where $W$ is the weight of fine particles, $S$ is surface area of sample plot.

A simplified Mass Balance Model is widely used for assessment of erosion rates on cultivated lands (Zhang et al., 1990):

$$A=A_0(1-h/H)^{N-1963} \tag{3}$$

Where $A_0$ is the $^{137}$Cs reference inventory (Bq m$^{-2}$); $A$ is the $^{137}$Cs inventory at an erosion point (Bq m$^{-2}$); $h$ is the annual soil

loss in depth since the year 1963 (cm); $H$ represents the plough depth (cm); and $N$ means the sampling year.

$^{137}$Cs has been widely used for dating of undisturbed soil profiles. The excepted $^{137}$Cs depth profile characterized by a single

peak for the year 1963 which the $^{137}$Cs maximum fallout flux occurred. The deposition rate since 1963 can be evaluated by

using the following equation (Bai et al., 2010):

$$R=H_m/(n-1963) \tag{4}$$

Where, $R$ is the deposition rate (cm a$^{-1}$), $H_m$ is the depth of the peak in $^{137}$Cs activity (cm), and $n$ is the sampling year.

The deposited sediments used to be mixed into a plough layer by plough activities at the karst depression bottom land. The $^{137}$Cs

distribution depth at the deposited depression bottom is greater than the local reference inventory. Under the assumption

that $^{137}$Cs fallout were totally on the ground in 1963, the sediment deposition depth was derived from the equation (Bai et al.,

140  2010):

$$\Delta H=H_m-H_p \tag{5}$$

Where, $\Delta H$ is the sediment deposition depth since 1963 (cm); $H_m$ represents the total $^{137}$Cs distribution in profile (cm); $H_p$

means the plough layer depth (cm).

The modulus of the soil erosion is calculated as follows:

$$Y=h \times D \times 10000 \tag{6}$$

$Y$ is soil erosion modulus (t km$^{-2}$a$^{-1}$); $D$ is the soil capacity (g cm$^{-3}$).





$R=1-Q_d/Q_m$ (7)

$R$ is sediment delivery ratio; $Q_d$ is the deposition amount(t); $Q_m$ is the erosion amount(t)

**2.5 Statistical analysis**

Data on soil pH, SOM, TN, TP and TK were statistically analysed to provide annual averages from each soil depths under different hillslope positions (shoulder, backslope and footslope). First, analysis of variance (ANOVA) was conducted to soil properties and sediment deposition rates, in order to evaluate their statistical differences at different positions and depths. In the case of obtaining significant differences at $P < 0.05$, the variable means were compared using an LSD (least significant difference) test. Additionally, a Principal Component Analysis (PCA) was also performed to determine first correlations among

the measured variables. The raw datasets were standardized before analyses and all statistical analyses were conducted using the R software version 3.2.4 (R Core Team 2013).

**3. Results**

**3.1 Variation of $^{137}$Cs and soil physicochemical properties at different hillslope position**

Our results showed that $^{137}$Cs concentration were significantly different at different hillslope positions (P<0.05) (Fig. 2). The

average $^{137}$Cs concentration was the highest in the backslope (0.83 Bq kg$^{-1}$), followed by the footslope (0.58 Bq kg$^{-1}$) and shoulder (0.20 Bq kg$^{-1}$). The $^{137}$Cs inventories at different hillslope positions were respectively 364.6 Bq m$^{-2}$, 249.9 Bq m$^{-2}$and 85.1 Bq m$^{-2}$, and the mean $^{137}$Cs inventories was 226.5 Bq m$^{-2}$.

Similar to the $^{137}$Cs concentration, we found the maximum values of soil pH, SOC, TN, TP, TK in the backslope. In addition, it is important to remark that soil pH, TN, TK were higher in the shoulder than in the footslopes, meanwhile, SOC and TP were

higher in the lower parts than in the upper ones. Except for SOC, other soil properties (pH, TN, TP and TK) were significantly different at different hillslope positions (Table 1).




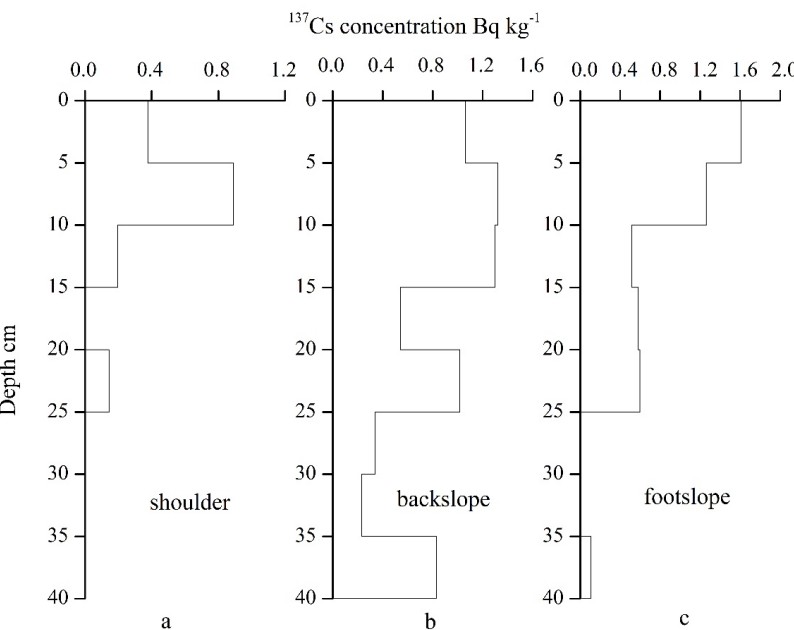

**Figure 2: $^{137}$Cs concentration distribution features in different position of hillslope.**


**Table 1: Variations in $^{137}$Cs and soil properties at different hillslpe positions.**

|  | $^{137}$Cs<br>Bq kg$^{-1}$ | pH | SOM<br>g kg$^{-1}$ | TN<br>g kg$^{-1}$ | TP<br>g kg$^{-1}$ | TK<br>g kg$^{-1}$ |
|---|---|---|---|---|---|---|
| Shoulder | 0.20±0.33b | 6.87±0.37b | 0.95±0.72a | 0.09±0.04b | 0.02±0.00b | 1.06±0.21b |
| Backslope | 0.82±0.65a | 7.47±0.74a | 1.27±0.52a | 0.14±0.05a | 0.03±0.01a | 1.95±0.54a |
| Footslope | 0.58±0.62a | 6.09±1.03c | 1.05±0.32a | 0.09±0.03b | 0.03±0.01b | 0.95±0.34b |

Data represent means and standard deviations (SD). Different lowercase letters indicate a significant difference among slope position.

### 3.2 Variation of $^{137}$Cs and soil physicochemical properties for the selected hillslopes at different soil depths

Figure 3 and Table 2 shows the variations of $^{137}$Cs and soil properties for the selected hillslopes and different soil depths, respectively. In the shoulder, $^{137}$Cs was mainly distributed in the topsoil (i.e., 0.38 Bq kg$^{-1}$ in 0-5 cm and 0.89 Bq kg$^{-1}$ in 5-10 cm soil). Below 10 cm, $^{137}$Cs concentration decreased rapidly. There were no $^{137}$Cs in 15-20 cm, 25-30 cm, 30-35 cm and 35-40 cm soil depths.

In the backslope, $^{137}$Cs concentration ranged from 0.23 Bq kg$^{-1}$ to 1.32 Bq kg$^{-1}$. $^{137}$Cs was mainly distributed within 0-15 cm

soil depth (i.e., 1.06 Bq kg$^{-1}$ in 0-5 cm, 1.32 Bq kg$^{-1}$ in 5-10 cm and 1.30 Bq kg$^{-1}$ in 10-15 cm soil). The mean $^{137}$Cs concentration of the whole soil profile was 0.83 Bq kg$^{-1}$.

Finally, in the footslope, the $^{137}$Cs concentration was mainly distributed within 0-10 cm. The peak concentration of $^{137}$Cs was





in the top 5 cm with a concentration of 1.61 Bq kg$^{-1}$, which was also the maximum concentration of whole hillslope. From the top 5 cm, $^{137}$Cs concentration decreased with increasing soil depth. There were no $^{137}$Cs in the 25-30 cm and 30-35 cm soil depths. The mean $^{137}$Cs concentration in the foot slope was 0.58 Bq kg$^{-1}$.

Based on the variance analysis considering the different five soil depths (0-5 cm, 5-10 cm,10-20 cm, 20-30 cm, 30-40 cm), $^{137}$Cs concentration was significantly different (P<0.05). Multiple comparison showed $^{137}$Cs concentration in 0-5 cm was significantly higher than that below 10 cm.$^{137}$Cs concentration in 5-10 cm was significantly different with that in 10-20 cm, 20-30 cm and 30-40 cm, but it was not significant with the 0-5 cm. $^{137}$Cs concentration in 10-20 cm, 20-30 cm and 30-40 cm showed no significant difference.

Other soil properties (SOM, TN, TP) showed significant differences (P<0.05), meanwhile, TK and pH showed no significant differences among soil depths. SOM in 0-5 cm (1.89 g kg$^{-1}$) was significantly higher than other soil depths and, even, SOM in 5-10 cm (1.40 g kg$^{-1}$) was significantly higher than 20-30 cm (0.88 g kg$^{-1}$), and 30-40 cm (0.62 g kg$^{-1}$). TN in adjacent soil depths registered no significant differences. For example, TN in 0-5 cm (0.16 g kg$^{-1}$) was significantly higher than the soil layers below 10 cm, but had no significant difference with 5-10 cm. TP in 0-5 cm (0.033 g kg$^{-1}$) and 5-10 cm (0.035 g kg$^{-1}$) were significant higher than the soil depths below 20 cm. Soil TP had no significant difference between two adjacent soil depths.

**Table 2. Variations in $^{137}$Cs and soil properties at different soil depths.**

|  | $^{137}$Cs Bq kg$^{-1}$ | pH | SOM g kg$^{-1}$ | TN g kg$^{-1}$ | TP g kg$^{-1}$ | TK g kg$^{-1}$ |
|---|---|---|---|---|---|---|
| 0-5 cm | 1.01±0.67a | 6.31±1.02a | 1.89±0.87a | 0.16±0.05a | 0.033±0.007a | 1.37±0.63a |
| 5-10 cm | 1.16±0.41a | 6.49±1.06a | 1.40±0.32b | 0.13±0.03ab | 0.035±0.011a | 1.33±0.68a |
| 10-20 cm | 0.52±0.54b | 6.82±0.95a | 1.04±0.27bc | 0.10±0.04bc | 0.028±0.009ab | 1.33±0.62a |
| 20-30 cm | 0.35±0.39b | 7.06±0.87a | 0.88±0.33c | 0.09±0.04cd | 0.022±0.006bc | 1.26±0.60a |
| 30-40 cm | 0.09±0.14b | 5.73±2.58a | 0.62±0.34c | 0.06±0.04d | 0.018±0.009c | 1.07±0.61a |

Data represent means and standard deviations (SD). Different lowercase letters indicate a significant difference among different depth (P<0.05).

### 3.3 $^{137}$Cs variation of depression bottom in different soil depth

We observed a possible trend in $^{137}$Cs concentration (Figure 3): first increased with soil depth, and then decreased after 165 cm. The peak of $^{137}$Cs concentration was in the 165 cm soil layer (2.38 Bq kg$^{-1}$). The mean $^{137}$Cs concentration in the whole depression soil profile was 1.25 Bq kg$^{-1}$, higher than the $^{137}$Cs concentration of the tested hillslopes.



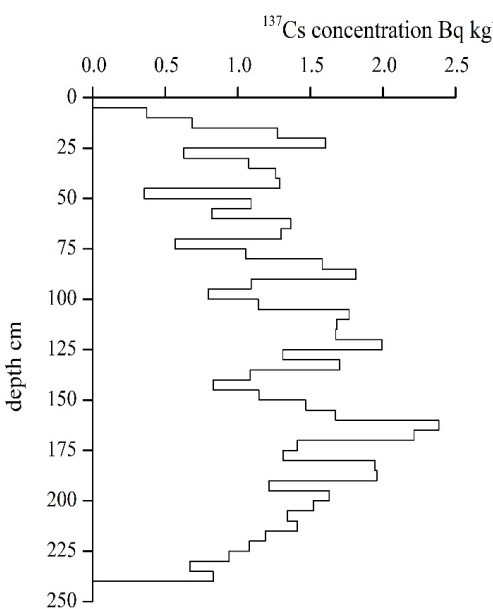

**Figure 3: <sup>137</sup>Cs depth distribution features in depression bottom.**

**Figure 3: $^{137}$Cs depth distribution features in depression bottom.**

**3.4 Potential connection between soil properties and sediment deposition rates**

210    A principal component analysis (PCA) was carried out considering the above-mentioned variables related to sediment
deposition rates using $^{137}$Cs. Fig. 4 showed a plot of the eigenvector in the plane of the first two components together with the
PC scores in the plane of PC1 and PC2. On the first component, which explained 61.4% of the total variance, and the second
component explained 23.6%, respectively. Sediment deposition rates (Cs) was significantly affected by hillslope positions,
meanwhile, Cs closely related to SOM and TN.





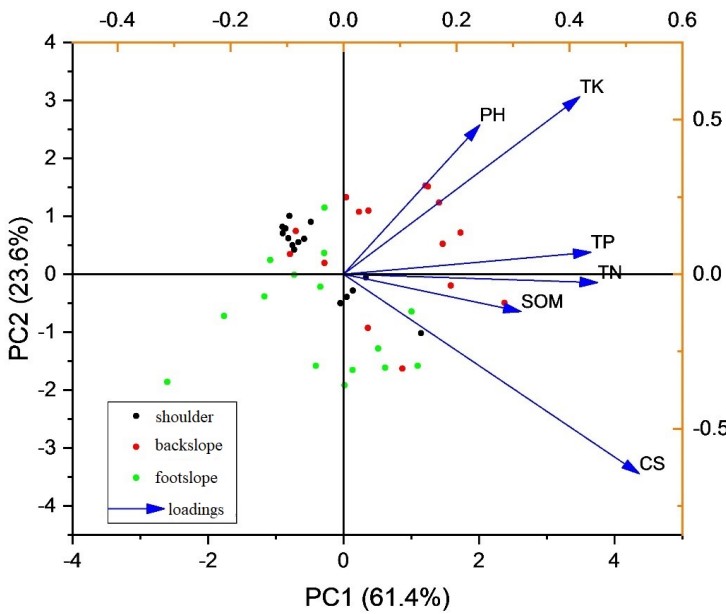

**Figure 4: Eigenvectors from the principal component analysis (PCA) of the first two components.**

**3.5 Soil erosion modulus estimation**

The mean $^{137}$Cs concentration in reference site was 6.28 Bq kg$^{-1}$. Using Eq. (2), we obtained 942 Bq m$^{-2}$ as the reference $^{137}$Cs inventory. The $^{137}$Cs inventory at different hillslope positions (shoulder, backslope, footslope) was lower than the reference plot inventory, possibly indicating that soil erosion happened in the disturbed hillslopes. Using the soil erosion rate Eq. (3) presented above, the calculated soil erosion rates in the shoulder, back- and footslopes were 0.87 cm a$^{-1}$, 0.35 cm a$^{-1}$ and 0.49 cm a$^{-1}$, respectively. Using the soil erosion modulus Eq. (6) and combining with the weight of different hillslope positions to calculate it, the average erosion modulus in the whole hillslope was 632 t km$^{-2}$a$^{-1}$.

In the depression bottom, the $^{137}$Cs distribution was much deeper than that of local $^{137}$Cs reference depth and the plough layer. The $^{137}$Cs inventory was much greater than the $^{137}$Cs reference inventory. Based on the depression bottom's $^{137}$Cs peak concentration (165 cm soil layer), the deposition depth since 1963 was 145 cm because the local plough layer is about 20 cm. From Eq. (4) and Eq. (5), we obtained that the average deposition rates since 1963 was 2.65 cm a$^{-1}$. Finally, the soil deposition modulus was 3180 t km$^{-2}$a$^{-1}$, calculated from Eq. (6).

**4. Discussion**

Choosing a reference plot was a critical step for assessing the soil erosion rates using $^{137}$Cs fingerprinting technique. While some researchers doubted this method because of $^{137}$Cs fallout heterogeneity (Parsons and Foster, 2011), the reference inventory in our study site was 942 Bq m$^{-2}$, close to the values of previous studies under similar rainfall and latitude conditions. For example, Zhang et al. (2009) used 918 Bq m$^{-2}$ as the ideal reference inventory plot in Dianchi watershed Yunnan province.





Similarly, Xiong et al (2018) used 906 Bq m$^{-2}$ as the reference inventory in Shilin county, Yunnan province.

While soil erosion is usually greatest in the shoulder, followed by the backslope and lowest in the footslope (Song et al., 2018), in our study, soil erosion was greater in either upper and lower hillslope parts than that in the middle one. Steeper gradient of the upper slope compared to the middle could be responsible for this finding. Previous study shows that slope gradient is the key factor affecting soil erosion when the rainfall and vegetation coverage are remaining unchanged (Lu et al., 2016). In backslope, chemical dissolution is strong, forming some relatively closed microtopography, such as lapies and solution pans.

Soil from the shoulder is easily deposited in these microtopographical forms (Zhang et al.,2009b). However, other research carried out in steep and conventionally tilled managed fields also highlighted the importance of the soil management practices conducted by handmade tillage or tractor passes redistributing soil sediments along the hillslope. This issue is closely related to the connectivity processes, which using this method could be uniquely estimated but not directly quantified. Therefore, in the future, using other techniques this factor should further studied and its influence deciphered.

In accordance with the trend in soil erosion observed in our results, $^{137}$Cs concentration peaked in 5-10 cm soil depths in the shoulder and footslope, but only in the first 5 cm soil depth of the backslope. These results implied that soil erosion is highly affecting the topsoil layers, whichis a serious issue, specifically, in the shoulder and footslope. On the contrary, the microtopography that is formed in the backslope could capture the soil carried by the erosion from the upper part, which could be responsible for elevated $^{137}$Cs concentration in the soil surface. However, this issue must be also studied in the future with

much more detail.

Our research also showed the $^{137}$Cs concentration of different soil depth was significant correlated with SOM. These results are consistent with previous studies conducted different environmental conditions (Parsons and Foster, 2011). This connection has a significant impact on the pathways of $^{137}$Cs movement in the near-surface environment (Agapkina et al., 1995). In soil profiles, the concentration of $^{137}$Cs show a discontinuous trace distribution in different depths. On one hand, it can be

hypothesized that there would be the possibility to observe soil creeping in soil and rock interfaces attending the parent material composition, in this case, carbonate rocks. On the other hand, $^{137}$Cs is mainly adsorbed by the fine soil particles and soil organic matter. The fine soil particles are easily to be mobilized to the deep soil layers under the action of gravity and leaching of rainwater.

The soil erosion modulus averaged 632 t/(km$^2$·a), which was much higher than the karst peak-cluster area in Longhe village

Guangxi Province (Luo et al., 2018). On the one hand, some treatments and preventions of soil erosion have made great progress and the soil erosion modulus decreased about 500 t km$^{-2}$ from 2003 to 2015. On the other hand, the soil in peak-cluster was thinner and some areas have no soils, just remaining bedrocks owing to previous soil erosion epochs. To understand the extent of soil erosion in our study site, we used the standard procedures established by Cao et al (2004) who considered the carbonate rock soil formation rate by analyzing the factors influencing soil formation. Cao et al (2004) used soil formation



rates as soil loss tolerance and redefined the classification standard of the intensity of soil and water losses in the karst area

into the following categories: very slight, slight, medium, serious and very serious (<30, 30-100, 100-200, 200-500, 500-1000

and >1000 t km$^{-2}$ a$^{-1}$, respectively). Based on the above-mentioned classification, our research site was seriously eroded, which

would require corrective measures to reduce soil erosion.

In our research site, the area of whole hillslope and depression bottom is 1.91 km$^2$ and 0.067 km$^2$, respectively Considering

the different soil erosion rate and weight of area at different hillslope positions, the annual soil loss was about 1207 t. According

to the sediment rates and depression bottom areas, the annual soil sediment reached only 213 t. In the same closed catchment,

the loss soil is much more than the sediment soil. This indicated that a part of the soil loss from the hillslope could be got into

subterranean stream from sinkhole of northeast depression. From the erosion and sediment difference, we obtained that the

sediment delivery ration in our study site was about 0.82. This result is greater than the empirical value of reasonable sediment

delivery ratio (0.7), which was obtained by Zhang et al (2010) using DEPOSITS model. Ward et al. (1981) proposed

DEPOSITS model for estimating sediment delivery ratio in small impoundments based on plug flow theory. The DEPOSITS

model assumes that sediment delivery ratio is in proportion to runoff retention time. In karst area, depression can be considered

as temporary impoundment because the area is easily flooded (Wang et al., 2004). At the same time, plug flow theory only

considers flooded conditions. Nowadays, in our study site, there are many rainfall events which do not lead to flooding.

Therefore, calculating sediment delivery ratio that considers local conditions could be more reliable than the generic empirical

value.

The research depression is the typical geomorphic unit of karst garbin basin. There are a great number of depressions in karst

garbin basin which are the major soil erosion area. When the soil erosion happens in the studied hillslope after a rainfall event,

the soil loss is divided into two parts according the sediment delivery ratio. 18% of soil loss is deposited in the depression

bottom, and 82% flowed into the subterranean stream system. The sediments which flowed into subterranean stream is also

divided into two parts. One part is deposited in the subterranean conduit; another is discharged from subterranean stream outlet.

The ratio of sediment in subterranean is the challenging and meaning research of next step.

Because soil erosion mainly happens on hillslopes which remain cultivated, biological measures such as afforestation using

deep-rooted species should be the prioritized. To encourage the shift in crop selection, high-value perennial crops such as

honeysuckle can be promoted. The use of plough should be avoided to further reduce soil erosion. Finally, growing plant fence

or sediment storage dam can be built in front of the sinkhole because most of lost soil flow into the sinkhole of depression

bottom.

**5. Conclusions**

The $^{137}$Cs are mainly distributed in the topsoil between 0 and 10 cm owing to the $^{137}$Cs adsorbed in the soil fine particles and

soil organic matter. On the shoulder and backslope, the maximum of $^{137}$Cs appears in the soil subsurface layers, whereas at the

footslope, the maximum appears in the topsoil parts. For this study case, soil erosion rate is 0.87 cm a$^{-1}$, 0.35 cm a$^{-1}$, 0.49 cm



a$^{-1}$ in the shoulder, back- and footslope, respectively. Possibly, the main driver factors werethe microtopographical cahnges and the strong slope gradients. Based on the sedimentation rates of the depression bottom (2.65 cm a$^{-1}$), sediment delivery ratio (0.82) and the average soil erosion modulus at each hillslope position (632 t km$^{-2}$a$^{-1}$), soil erosion intensity can be considered

according to the karst soil erosion gradation criterion. In order to reduce soil erosion in this region, we suggest some control measures that are feasible and can be implemented, even by farmers.

**Acknowledgments**

This work is jointly financially supported by National Key Research and Development Program (Evolution, integrating

treatment and technological demonstration of rocky desertification in Karst Grabin Basin. Grant No.2016YFC0502503), Guangxi Natural Science Foundation (Grant No. 2017GXNSFBA198037; 2017JJA150639y) and the National Natural Science Foundation of China (Grant No. 41502342). Young Elite Scientist Sponsorship Program by the China Association for Science and Technology (2017–2019, awarded to Dr. Yang Yu). Contributions from Dr. Ruirui Cheng and Dr. Fan Liu from Institute of Karst Geology and many others are greatly appreciated.

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
