# Peer review of "Evaluating soil erosion and sediment deposition rates by the 137Cs fingerprinting technique at karst gabin basin in Yunnan Province, southwest China"

_SOIL, 2019_

## Referee Comment (RC1) · Anonymous Referee #1 · 11 Feb 2020

General comments Li et al., display an interesting study based on 137Cs activity and soil property measurements carried out in a karst depression in SW China to estimate soil erosion rates along a cultivated catchment's slope and related sediment accumulation rate in its bottom part. The authors only sampled 10 soil cores (nine along 3 hillslope positions and one within the depression). Estimates of soil erosion rates were derived from the soil 137Cs inventories for a limited number of sites using the method published by Zhang et al. (2009). They also carried out a PCA analysis to relate several soil properties (soil pH, total nitrogen - total phosphorus - total potassium

concentrations and soil organic matter content) with sediment deposition rates derived from 137Cs activity measurements. The study aims to provide information on land degradation due to soil particle's redistributions (mainly erosion) for policy makers and stakeholders. I think that the paper in its present form raises several major questions.

Specific comments 1) Estimates of soil particle's redistribution rate in a catchment require a reference 137Cs fallout level, estimated to be 942 Bq/m2 in this study. It is assumed that this reference site neither lost nor gained soil particles since the deposition maximum of 1963. Soil cores that display 137Cs inventories above or below this value are then interpreted as accumulation or erosion sites, respectively. Details of the calculation of this reference (average?) value are missing in the paper (i.e.137Cs activity distributions with soil depth, soil densities, plough depth, particle size,...). I think that this important information should be reported somewhere in the paper, together with some discussion with respect to a homogenous fallout. On Line 124 in section 2.4, it is mentioned "Reference sample was considered using a bulk sample..." but the 137Cs activities of the samples are determined on sieved <2 mm soil fractions. Both may not be comparable?

2) The authors assume that 137Cs accumulation peaked at the 165 cm soil depth (2.38 Bq kg-1) in the bottom part of the catchment (Fig. 3), providing a deposition rate of 2.65 cm yr-1 (and a soil accumulation of 3180 t km-2 yr-1, reported Line 228 by the authors). However another peak can be found just below at 190 cm with approximately the same value (ca. 2.0 Bq kg-1) than at 165 cm soil depth (taking into account the analytical uncertainty). Assuming the same deposition rate, the corresponding date would be 1954 (25 cm / 2,65 cm yr-1 corresponding to ca. 9 yr before 1963). This time period is rather known as the onset of 137Cs fallout than a high fallout deposition year. I think that there is a large uncertainty on the reference 1963 fallout peak position (somewhere between 150-200 cm soil depth?) possibly due to soil particle's mixing if land was cultivated or to a more complex deposition trend including a varying supply of 137Cs-tagged soil particles. Accordingly any deposition rate that can be derived using

this soil depth may be questioned.

3) In the discussion section, the authors mention, on the basis of their 137Cs inventories, that soil erosion is lower in the middle part of the hill slope than in the upper and lower positions (Lines 235-250). I suggest that the authors provide references to support this interpretation (i.e., Ribolzi et al. 2011 - Geomorphology 127, 53-63 or others). It is also worth noticing that a correlation between 137Cs activity and SOM content is assumed (Line 251-258). However the discussion is difficult to follow because the authors do no plot any correlations, only a PCA analysis showing "trends" between soil properties (Fig. 4). I think that graphical plots (i.e., SOM content vs. 137Cs activity in concentration units and/or SOM kg m-2 vs. 137Cs in Bq.m-2) could help the reader to better evaluate the "reality of things". I think that if such a correlation exists it may not be directly due to 137Cs adsorption by soil organic matter (Line 256) but rather to the fact that soil micro-aggregates contain both organic matter and 137Cs bound to fine clay minerals. On the long term a single process, i.e. erosion, will deplete topsoil horizons in both soil organic matter and particle's bound 137Cs during soil aggregate breakdown.

Technical comments I think that some improvements should be made for the figures and tables. Line 175 it is mentioned Fig.3 but I think it should rather be Fig.2. Moreover In Fig.2 the reader does not know if average or single values are plotted? In the case of average values, how many values (3 for the 3 soil cores)? Nothing is said about this in the legend. In such a case the SD should also be reported in Fig.2. The title of Table 2 "Variations in 137Cs and soil properties..." might rather be "Average variations in..." ?

---

## Referee Comment (RC2) · Anonymous Referee #2 · 19 Feb 2020

This paper aims to quantify the erosion and sediment deposition rates in the Karst region of Southwest China using 137Cs tracing technique. Further, the authors evaluated the relationships between 137Cs and selected soil properties (soil pH, total nitrogen, total phosphorus, and SOC content) by PCA analysis. The purpose of this study is worth giving the intensity of soil erosion in the area. However, I have many concerns about the conclusions:

1. The authors only sampled 10 soil cores (nine along 3 transects and one from the depression). I think the size of samples is inadequate for obtaining a catchment-scale

conclusion, e.g. Line 20 "the sediment delivery ratio summarized 0.82 in the whole catchment according to the square of hillslope and depression bottom". Given the complexity of topography of the study area showing in Fig. 1, erosion rates and soil properties can be highly variable.

2. Statistic relationship between 137Cs and soil properties cannot be obtained by PCA (Line 25). The angle between two variables in PAC Biplot just indicates a tendency of correlation. The authors should perform simple correlation analysis to confirm the statistic results. Further, PCA is a technique for reducing the dimensionality of complex datasets, increasing interpretability. I can't see the necessity to perform PCA in this paper in its present form. I would suggest authors try to 1) explain the first two components 2) combine PCA with PERMANOVA to examine how do measured variables differ between slope positions. Then reconsider the necessity of using PCA.

3. §3.2 Authors presented the variation of 137Cs and soil physicochemical properties for selected hillslope at different soil depths. It is not clear which slope position you selected for such comparison? Only shoulder position or plus foot slope? Why?

4. §5 In conclusions (Line 295), authors mentioned that "on the shoulder and backslope, the maximum of 137Cs appears in the soil subsurface layers, whereas at the footslope, the maximum appears in the topsoil parts ". I will doubt this conclusion unless the SD value can be reported in Fig. 2. Its important because what I can see from Fig. 2 is that there might be no difference (if high SD) between 0-5 cm (topsoil) and 5-10 cm (subsurface) at backslope and footslope. From my point of view, it's reasonable that no difference of 137Cs values between 0-5 cm and 5-10 cm because of the mixed effect of tillage practice.

5. Please report the slope gradients in Table 2, then we can see the rationality of your explanations for the factors driving erosion rates (§Conclusions, Line 297).

6. This paper showed that soil erosion was greater in either upper and lower hillslope parts than in the middle one (Line 235), and authors attribute these patterns to the

slope gradient. I think another possible reason is that the coexisting of tillage erosion and water erosion. Typically, tillage erosion is the main cause of soil loss at the concave position (ref. to Lobb D.A. 1999), i.e. shoulder position (upper parts), while water erosion leads to serious soil loss at lower slope position (these areas received maximum runoff concentrations).

Technical comments:

7. Line 23: ". . . . . .play the most important role in WHAT?

8. Line 113: is there inorganic C from the soil samples? If so, how did you remove it?

9. Line 160: add SD to 137Cs concentration.

10. Line 175: Fig.2 rather than Fig.3?

11. Line 242: please add a reference

12. Line 245-246: what you mentioned here is not correct according to Fig. 2. Please check it carefully.

13. Line 261: please add ref. here to show where is this data from

14. Line 297: check spell of letters.

---

## Author Comment (AC1) · 6 Mar 2020

Revision Note for Reviewer 1

General comments Li et al., display an interesting study based on 137Cs activity and soil property measurements carried out in a karst depression in SW China to estimate soil erosion rates along a cultivated catchment's slope and related sediment accumulation rate in its bottom part. Dear reviewer 1, thanks a lot for your time invested in our manuscript. We highly appreciate your comments and suggestions. We tried to do our

best in order to improve our research.

The authors only sampled 10 soil cores (nine along 3 hillslope positions and one within the depression). Estimates of soil erosion rates were derived from the soil 137Cs inventories for a limited number of sites using the method published by Zhang et al. (2009). Yes, to obtain our final conclusions and achieve our goals, we considered that the number of samples are representative. However, we will include this possible issue for you in the discussion part, in order to make clearer your point of view and ours for the readers.

They also carried out a PCA analysis to re-late several soil properties (soil pH, total nitrogen - total phosphorus - total potassium concentrations and soil organic matter content) with sediment deposition rates derived from 137Cs activity measurements. The study aims to provide information on land degradation due to soil particle's redistributions (mainly erosion) for policy makers and stakeholders. I think that the paper in its present form raises several major questions. Yes, we agree with you and are happy to see that all your comments are very valuable to improve our ms. We are also happy to see that you found interesting the main goal of our paper, which is vital to protect our environment.

Specific comments 1) Estimates of soil particle's redistribution rate in a catchment require a reference 137Cs fallout level, estimated to be 942 Bq/m2 in this study. It is assumed that this reference site neither lost nor gained soil particles since the deposition maximum of 1963. Soil cores that display 137Cs inventories above or below this value are then interpreted as accumulation or erosion sites, respectively. Details of the calculation of this reference (average?) value are missing in the paper (i.e.137Cs activity distributions with soil depth, soil densities, plough depth, particle size,...). I think that this important information should be reported somewhere in the paper, together with some discussion with respect to a homogenous fallout. On Line 124 in section 2.4, it is mentioned "Reference sample was considered using a bulk sample..." but the 137Cs activities of the samples are determined on sieved <2 mm soil fractions. Both

may not be comparable?

Response: Thank you very much for your comments. The reference value is important. We revised that mentioned section. Please, see the 3.5 sub-chapter in the revision manuscript. We also added some discussion about homogenous fallout in the discussion part. Yes, sorry for the non-clear explanation. The bulk sample corresponded to the sample without layered. Both are not comparable. All the samples were sieved 2 mm including the reference sample before measuring 137Cs.

2) The authors assume that 137Cs accumulation peaked at the 165 cm soil depth(2.38 Bq kg-1) in the bottom part of the catchment (Fig. 3), providing a deposition rate of 2.65 cm yr-1 (and a soil accumulation of 3180 t km-2 yr-1, reported Line 228 by the authors). However another peak can be found just below at 190 cm with approximately the same value (ca. 2.0 Bq kg-1) than at 165 cm soil depth (taking into account the analytical uncertainty). Assuming the same deposition rate, the corresponding date would be 1954 (25 cm / 2,65 cm yr-1 corresponding to ca. 9 yr before 1963). This time period is rather known as the onset of 137Cs fallout than a high fallout deposition year. I think that there is a large uncertainty on the reference 1963 fallout peak position (somewhere between 150-200 cm soil depth?) possibly due to soil particle's mixing if land was cultivated or to a more complex deposition trend including a varying supply of 137Cs-tagged soil particles. Accordingly any deposition rate that can be derived using this soil depth may be questioned.

Response: Thank you very much for your comments. We included these interesting ideas in our discussion. We consider that they are vital to improve our paper. The method using 137Cs concentration to calculate the soil deposition rate is a usual way in karst depressions (Bai et al,2010; Zhang et al, 2010, and so on). 137Cs is an artificial radionuclide released as a result of atmospheric testing during 1954 to the 1965. The maximum deposition rate was in 1963-1964 in the northern hemisphere. So, we consider that it is correct the mention of two possible peaks. The highest peak stands for 1963 and another one for 1954. Sorry for our calculation, there is a small

mistake, not 2.65 cm yr-1 but 2.68 cm yr-1, we revised it.

3) In the discussion section, the authors mention, on the basis of their 137Cs inventories, that soil erosion is lower in the middle part of the hill slope than in the upper and lower positions (Lines 235-250). I suggest that the authors provide references to sup-port this interpretation (i.e., Ribolzi et al. 2011 - Geomorphology 127, 53-63 or others).It is also worth noticing that a correlation between 137Cs activity and SOM contentis assumed (Line 251-258). However the discussion is difficult to follow because the authors do no plot any correlations, only a PCA analysis showing "trends" between soil properties (Fig. 4). I think that graphical plots (i.e., SOM content vs. 137Cs activity in concentration units and/or SOM kg m-2 vs. 137Cs in Bq.m-2) could help the reader to better evaluate the "reality of things". I think that if such a correlation exists it may not be directly due to 137Cs adsorption by soil organic matter (Line 256) but rather to the fact that soil micro-aggregates contain both organic matter and 137Cs bound to fine clay minerals. On the long term a single process, i.e. erosion, will deplete topsoil horizons in both soil organic matter and particle's bound 137Cs during soil aggregate breakdown.

Response: Thank you very much for your comments. We think that this suggestion deserves to be included using similar words like you mentioned. If you agree, we included this in our discussion section. In line 235-250, we add the references to confirm these conclusions and support the interpretation. And we also add the correlation between 137Cs and SOM and your correct interpretation. Thanks a lot.

Technical comments I think that some improvements should be made for the figures and tables. Line 175 it is mentioned Fig.3 but I think it should rather be Fig.2. Moreover In Fig.2 the reader does not know if average or single values are plotted? In the case of average values, how many values (3 for the 3 soil cores)? Nothing is said about this in the legend. In such a case the SD should also be reported in Fig.2. The title of Table 2 "Variations in 137Cs and soil properties..." might rather be "Average variations in..." Response: Thank you very much for your comments. Line175 is Fig.2., yes we

revised it. Fig.2 include the average values and they are plotted. We redraw the plots and added the SD in Fig.2. The title of Table 2 was also revised.

Please also note the supplement to this comment:
https://www.soil-discuss.net/soil-2019-94/soil-2019-94-AC1-supplement.pdf

[Figure]

Figure 1: Localisation of the study area, sampling points and panoramic image of one selected plot.

**Fig. 1.**

[Figure]

Figure 2 : ¹³⁷Cs concentration distribution features at different hillslope positions.

**Fig. 2.**

¹³⁷Cs depth distribution figure

Figure 3: ¹³⁷Cs depth distribution features in depression bottom.

**Fig. 3.**

[Figure]

**Figure 4: Eigenvectors from the principal component analysis (PCA) of the first two components.**

**Fig. 4.**

Y=1.02x-0.33
r=0.669

Figure 5: Linear correlation between [137]Cs and SOM (Soil Organic Matter) content.

**Fig. 5.**

**Supplement:**

March 6th, 2019 Editor-in-Chief SOIL Dear Editor and Reviewers:

First of all, please let us transfer our most sincere thanks to your valuable comments for our manuscript. After receive the suggestions and comments, we have checked the whole paper thoroughly and revised it according to the constructive comments. The comments pointed out by the reviewers have been carefully treated. I hope all our efforts can make this paper more suitable for the publication level.

We uploaded the revision notes for the reviewers and the new version of the manuscript.

Thank you for your consideration and I looking forward to the further comments of reviewers to our study.

Sincerely,

Yang Yu theodoreyy@gmail.com Department of Sediment Research, China Institute of Water Resources and Hydropower Research

**Revision Note for Reviewer 1**

General comments Li et al., display an interesting study based on 137Cs activity and soil property measurements carried out in a karst depression in SW China to estimate soil erosion rates along a cultivated catchment's slope and related sediment accumulation rate in its bottom part. Dear reviewer 1, thanks a lot for your time invested in our manuscript. We highly appreciate your comments and suggestions. We tried to do our best in order to improve our research.

The authors only sampled 10 soil cores (nine along 3 hillslope positions and one within the depression). Estimates of soil erosion rates were derived from the soil 137Cs inventories for a limited number of sites using the method published by Zhang et al. (2009).

Yes, to obtain our final conclusions and achieve our goals, we considered that the number of samples are representative. However, we will include this possible issue for you in the discussion part, in order to make clearer your point of view and ours for the readers.

They also carried out a PCA analysis to re-late several soil properties (soil pH, total nitrogen - total phosphorus - total potassium concentrations and soil organic matter content) with sediment deposition rates derived from 137Cs activity measurements. The study aims to provide information on land degradation due to soil particle's redistributions (mainly erosion) for policy makers and stakeholders. I think that the paper in its present form raises several major questions.

Yes, we agree with you and are happy to see that all your comments are very valuable to improve our ms. We are also happy to see that you found interesting the main goal of our paper, which is vital to protect our environment.

Specific comments 1) Estimates of soil particle's redistribution rate in a catchment require a reference 137Cs fallout level, estimated to be 942 Bq/m2 in this study. It is assumed that this reference site neither lost nor gained soil particles since the deposition maximum of 1963. Soil cores that display 137Cs inventories above or below this value are then interpreted as accumulation or erosion sites, respectively. Details of the calculation of this reference (average?) value are missing in the paper (i.e.137Cs activity distributions with soil depth, soil densities, plough depth, particle size,...). I think that this important information should be reported somewhere in the paper, together with some discussion with respect to a homogenous fallout. On Line 124 in section 2.4, it is mentioned "Reference sample was considered using a bulk sample..." but the 137Cs activities of the samples are determined on sieved <2 mm soil fractions. Both may not be comparable?

Response: Thank you very much for your comments. The reference value is important. We revised that mentioned section. Please, see the 3.5 sub-chapter in the revision manuscript. We also added some discussion about homogenous fallout in the discussion part.

Yes, sorry for the non-clear explanation. The bulk sample corresponded to the sample without layered. Both are not comparable. All the samples were sieved 2 mm including the reference sample before measuring 137Cs.

2) The authors assume that 137Cs accumulation peaked at the 165 cm soil depth(2.38 Bq kg-1) in the bottom part of the catchment (Fig. 3), providing a deposition rate of 2.65 cm yr-1 (and a soil accumulation of 3180 t km-2 yr-1, reported Line 228 by the authors). However another peak can be found just below at 190 cm with approximately the same value (ca. 2.0 Bq kg-1) than at 165 cm soil depth (taking into account the analytical uncertainty). Assuming the same deposition rate, the corresponding date would be 1954 (25 cm / 2,65 cm yr-1 corresponding to ca. 9 yr before 1963). This time period is rather known as the onset of 137Cs fallout than a high fallout deposition year. I think that there is a large uncertainty on the reference 1963 fallout peak position (somewhere between 150-200 cm soil depth?) possibly due to soil particle's mixing if land was cultivated or to a more complex deposition rate that can be derived using this soil depth may be questioned.

Response: Thank you very much for your comments. We included these interesting ideas in our discussion. We consider that they are vital to improve our paper. The method using 137Cs concentration to calculate the soil deposition rate is a usual way in karst depressions (Bai et al,2010; Zhang et al, 2010, and so on). 137Cs is an artificial radionuclide released as a result of atmospheric testing during 1954 to the 1965. The maximum deposition rate was in 1963-1964 in northern hemisphere. So, we consider that it is correct the mention of two possible peaks. The highest peak stands for 1963 and another one for 1954. Sorry for our calculation, there is a small mistake, not 2.65 cm yr-1 but 2.68 cm yr-1, we revised it.

3) In the discussion section, the authors mention, on the basis of their 137Cs invento-ries, that soil erosion is lower in the middle part of the hill slope than in the upper andlower positions (Lines 235-250). I suggest that the authors provide references to sup-port this interpretation (i.e., Ribolzi et al. 2011 - Geomorphology 127, 53-63 or others). It is also worth noticing that a correlation between 137Cs activity and SOM contentis assumed (Line 251-258). However the discussion is difficult to follow because the authors do no plot any correlations, only a PCA analysis showing "trends" between soil properties (Fig. 4). I think that graphical plots (i.e., SOM content vs. 137Cs activity in concentration units and/or SOM kg m-2 vs. 137Cs in Bq.m-2) could help the reader to better evaluate the "reality of things". I think that if such a correlation exists it may not be directly due to 137Cs adsorption by soil organic matter (Line 256) but rather to the fact that soil micro-aggregates contain both organic matter and 137Cs bound to fine clay minerals. On the long term a single process, i.e. erosion, will deplete topsoil horizons in both soil organic matter and particle's bound 137Cs during soil aggregate breakdown.

Response: Thank you very much for your comments. We think that this suggestion deserves to be included using similar words like you mentioned. If you agree, we included this in our discussion section. In line 235-250, we add the references to confirm these conclusions and support the interpretation. And we also add the correlation between 137Cs and SOM and your correct interpretation. Thanks a lot.

Technical comments I think that some improvements should be made for the figures and tables. Line 175 it is mentioned Fig.3 but I think it should rather be Fig.2. Moreover In Fig.2 the reader does not

know if average or single values are plotted? In the case of average values, how many values (3 for the 3 soil cores)? Nothing is said about this in the legend. In such a case the SD should also be reported in Fig.2. The title of Table 2 "Variations in 137Cs and soil properties..." might rather be "Average variations in..."

Response: Thank you very much for your comments. Line175 is Fig.2., yes we revised it. Fig.2 include the average values and they are plotted. We redraw the plots and added the SD in Fig.2. The title of Table 2 was also revised.

**Evaluating soil erosion and sediment deposition rates by the 137Cs fingerprinting technique at different hillslope positions to assess a karst gabin basin in Yunnan Province, southwest China**

Yanqing Li1,2,3, Zhongcheng Jiang2,3, Zhihua Chen1, Yang Yu4\*, Funing Lan2,3, Xiangfei Yue2,3, Peng Liu2,3 Jesús Rodrigo-Comino5,6

1 China University of Geosciences, Wuhan 430074, China

[revised manuscript text omitted]

**2 Materials and methods**

**2.1 Study area**

The Dapotou depression is located in the Yangjie Town, Kaiyuan County, situated in the Yunnan Province from China (103°
17' 25.63"-103°18' 3.40" E, 23° 36' 48.04"-23° 37' 28.10"N) (Figure 1). This territory is a closed watershed with a drainage area of about 1.97 km2. Its elevations range from 1267 to 1413 m a.s.l. The underlying bedrock of the depression is a Triassic carbonate rock, which consists of Gejiu Group (T2g3) and Falang Group (T2f1) limestone. Most soils in this watershed have a soil texture of clay-limestone materials.

75

80

---

## Author Comment (AC2) · 6 Mar 2020

Revision Note for Reviewer 2

This paper aims to quantify the erosion and sediment deposition rates in the Karst region of Southwest China using 137Cs tracing technique. Further, the authors evaluated the relationships between 137Cs and selected soil properties (soil pH, total nitrogen, total phosphorus, and SOC content) by PCA analysis. The purpose of this study is worth giving the intensity of soil erosion in the area. However, I have many concerns

about the conclusions: Dear reviewer 2, also, thanks a lot for your time invested in our manuscript. We highly appreciate your comments and suggestions. We tried to do our best in order to improve our research and clarify all your concerns.

1. The authors only sampled 10 soil cores (nine along 3 transects and one from the depression). I think the size of samples is inadequate for obtaining a catchment-scale conclusion, e.g. Line 20 "the sediment delivery ratio summarized 0.82 in the whole catchment according to the square of hillslope and depression bottom". Given the complexity of topography of the study area showing in Fig. 1, erosion rates and soil properties can be highly variable.

Response: Thank you very much for your comments. Of course, understanding the high heterogeneity of soils need an elevated number of plot sites to increase the precision. The depression we choose is a small closed watershed. We chose different hillslope positions for sampling in order to decrease the heterogeneity. We consider that it can be adequate to get the small catchment-scale conclusion as other authors also found karst areas (Bai et al, 2010; Zhang et al, 2010, etc.). However, we included this idea as also suggested the reviewer 1.

2. Statistic relationship between 137Cs and soil properties cannot be obtained by PCA (Line 25). The angle between two variables in PAC Biplot just indicates a tendency of correlation. The authors should perform simple correlation analysis to confirm the statistic results. Further, PCA is a technique for reducing the dimensionality of complex datasets, increasing interpretability. I can't see the necessity to perform PCA in this paper in its present form. I would suggest authors try to 1) explain the first two components 2) combine PCA with PERMANOVA to examine how do measured variables differ between slope positions. Then reconsider the necessity of using PCA. Response: Thank you very much for your comments. We used an Anova for distinguishing the soil properties at different slope positions. We also performed the correlation analysis of the soil properties. However, we would want to conserve the use of the PCA in this paper, because it was really useful to observe and reduce the number of parameters able

to explain the interactions among soil processes at the hillslope scale. As we included a new analysis (linear correlation and Pearson correlation), we will wait for your new valuable evaluation to observe if you find now interesting this new approach. If not, we will delete it.

3. §3.2 Authors presented the variation of 137Cs and soil physicochemical properties for selected hillslope at different soil depths. It is not clear which slope position you selected for such comparison? Only shoulder position or plus foot slope? Why? Response: Thank you very much for your comments. Sorry, we did not express it clearly. The three hillslope positions were compared and now included in the text. We deleted the confused words.

4. §5 In conclusions (Line 295), authors mentioned that "on the shoulder and back-slope, the maximum of 137Cs appears in the soil subsurface layers, whereas at the footslope, the maximum appears in the topsoil parts ". I will doubt this conclusion unless the SD value can be reported in Fig. 2. Its important because what I can see from Fig. 2 is that there might be no difference (if high SD) between 0-5 cm (topsoil) and 5-10 cm (subsurface) at backslope and footslope. From my point of view, it's reasonable that no difference of 137Cs values between 0-5 cm and 5-10 cm because of the mixed effect of tillage practice. Response: Thank you very much for your comments. We added the SD values in Fig.2 because we agree with your valuable opinion. There are no differences of 137Cs values between 0-5 cm and 5-10 cm unless that we include more data such as SD.

5. Please report the slope gradients in Table 2, then we can see the rationality of your explanations for the factors driving erosion rates (, Line 297). Response: Thank you very much for your comments. We added the slope gradient in Table 2.

6. This paper showed that soil erosion was greater in either upper and lower hillslope parts than in the middle one (Line 235), and authors attribute these patterns to the slope gradient. I think another possible reason is that the coexisting of tillage erosion

and water erosion. Typically, tillage erosion is the main cause of soil loss at the concave position (ref. to Lobb D.A. 1999), i.e. shoulder position (upper parts), while water erosion leads to serious soil loss at lower slope position (these areas received maximum runoff concentrations).

Response: Thank you very much for your comments. It is a great idea! We included this in the discussion part. Thanks.

Technical comments: 7. Line 23: "......play the most important role in WHAT?

Response: "in influencing 137Cs".

8. Line 113: is there inorganic C from the soil samples? If so, how did you remove it?

Response: There is inorganic C. But our research only paid attention to soil organic matter.

9. Line 160: add SD to 137Cs concentration. Response: We added the SD.

10. Line 175: Fig.2 rather than Fig.3? Response: we revised it.

11. Line 242: please add a reference Response: We added.

12. Line 245-246: what you mentioned here is not correct according to Fig. 2. Please check it carefully. Response: Revised it.

13. Line 261: please add ref. here to show where is this data from Response: We added it.

14. Line 297: check spell of letters. Response: We checked and revised the letters.

Please also note the supplement to this comment:
https://www.soil-discuss.net/soil-2019-94/soil-2019-94-AC2-supplement.pdf

[Figure]

Figure 1: Localisation of the study area, sampling points and panoramic image of one selected plot.

**Fig. 1.**

[Figure]

Figure 2 : $^{137}$Cs concentration distribution features at different hillslope positions.

**Fig. 2.**

Figure 3: **¹³⁷Cs depth distribution features in depression bottom.**

**Fig. 3.**

[Figure]

**Figure 4: Eigenvectors from the principal component analysis (PCA) of the first two components.**

**Fig. 4.**

[Figure]

Figure 5: Linear correlation between ¹³⁷Cs and SOM (Soil Organic Matter) content.

**Fig. 5.**

**Supplement:**

March 6th, 2019
Editor-in-Chief
SOIL
Dear Editor and Reviewers:

First of all, please let us transfer our most sincere thanks to your valuable comments for our manuscript. After receive the suggestions and comments, we have checked the whole paper thoroughly and revised it according to the constructive comments. The comments pointed out by the reviewers have been carefully treated. I hope all our efforts can make this paper more suitable for the publication level.

We uploaded the revision notes for the reviewers and the new version of the manuscript.

Thank you for your consideration and I looking forward to the further comments of reviewers to our study.

Sincerely,

Yang Yu
theodoreyy@gmail.com
Department of Sediment Research,
China Institute of Water Resources and Hydropower Research

**Revision Note for Reviewer 2**

This paper aims to quantify the erosion and sediment deposition rates in the Karst region of Southwest China using 137Cs tracing technique. Further, the authors evaluated the relationships between 137Cs and selected soil properties (soil pH, total nitrogen, total phosphorus, and SOC content) by PCA analysis. The purpose of this study is worth giving the intensity of soil erosion in the area. However, I have many concerns about the conclusions:

**Dear reviewer 2, also, thanks a lot for your time invested in our manuscript. We highly appreciate your comments and suggestions. We tried to do our best in order to improve our research and clarify all your concerns.**

1. The authors only sampled 10 soil cores (nine along 3 transects and one from the depression). I think the size of samples is inadequate for obtaining a catchment-scale conclusion, e.g. Line 20 "the sediment delivery ratio summarized 0.82 in the whole catchment according to the square of hillslope and depression bottom". Given the complexity of topography of the study area showing in Fig. 1, erosion rates and soil properties can be highly variable.

Response: Thank you very much for your comments. Of course, understanding the high heterogeneity of soils need an elevated number of plot sites to increase the precision. The depression we choose is a small closed watershed. We chose different hillslope positions for sampling in order to decrease the heterogeneity. We consider that it can be adequate to get the small catchment-scale conclusion as other authors also found karst areas (Bai et al, 2010; Zhang et al, 2010, etc.). However, we included this idea as also suggested the reviewer 1.

2. Statistic relationship between 137Cs and soil properties cannot be obtained by PCA (Line 25). The angle between two variables in PAC Biplot just indicates a tendency of correlation. The authors should perform simple correlation analysis to confirm the statistic results. Further, PCA is a technique for reducing the dimensionality of complex datasets, increasing interpretability. I can't see the necessity to perform PCA in this paper in its present form. I would suggest authors try to 1) explain the first two components 2) combine PCA with PERMANOVA to examine how do measured variables differ between slope positions. Then reconsider the necessity of using PCA.

Response: Thank you very much for your comments. We used an Anova for distinguishing the soil properties at different slope positions. We also performed the correlation analysis of the soil properties. However, we would want to conserve the use of the PCA in this paper, because it was really useful to observe and reduce the number of parameters able to explain the interactions among soil processes at the hillslope scale. As we included a new analysis (linear correlation and Pearson correlation), we will wait for your new valuable evaluation to observe if you find now interesting this new approach. If not, we will delete it.

3. §3.2 Authors presented the variation of 137Cs and soil physicochemical properties for selected hillslope at different soil depths. It is not clear which slope position you selected for such comparison? Only shoulder position or plus foot slope? Why?

Response: Thank you very much for your comments. Sorry, we did   not express it clearly. The

three hillslope positions were compared and now included in the text. We deleted the confused words.

4. §5 In conclusions (Line 295), authors mentioned that "on the shoulder and backslope, the maximum of 137Cs appears in the soil subsurface layers, whereas at the footslope, the maximum appears in the topsoil parts ". I will doubt this conclusion unless the SD value can be reported in Fig. 2. Its important because what I can see from Fig. 2 is that there might be no difference (if high SD) between 0-5 cm (topsoil) and 5-10 cm (subsurface) at backslope and footslope. From my point of view, it's reasonable that no difference of 137Cs values between 0-5 cm and 5-10 cm because of the mixed effect of tillage practice.

Response: Thank you very much for your comments. We added the SD values in Fig.2 because we agree with your valuable opinion. There are no differences of 137Cs values between 0-5 cm and 5-10 cm unless that we include more data such as SD.

5. Please report the slope gradients in Table 2, then we can see the rationality of your explanations for the factors driving erosion rates (§Conclusions, Line 297).

Response: Thank you very much for your comments. We added the slope gradient in Table 2.

6. This paper showed that soil erosion was greater in either upper and lower hillslope parts than in the middle one (Line 235), and authors attribute these patterns to the slope gradient. I think another possible reason is that the coexisting of tillage erosion and water erosion. Typically, tillage erosion is the main cause of soil loss at the concave position (ref. to Lobb D.A. 1999), i.e. shoulder position (upper parts), while water erosion leads to serious soil loss at lower slope position (these areas received maximum runoff concentrations).

Response: Thank you very much for your comments. It is a great idea! We included this in the discussion part. Thanks.

Technical comments:
7. Line 23: "......play the most important role in WHAT?

Response: "in influencing $^{137}$Cs".

8. Line 113: is there inorganic C from the soil samples? If so, how did you remove it?

Response: There is inorganic C. But our research only paid attention to soil organic matter.

9. Line 160: add SD to 137Cs concentration.
   Response: We added the SD.

10. Line 175: Fig.2 rather than Fig.3?
Response: we revised it.

11. Line 242: please add a reference

Response: We added.

12. Line 245-246: what you mentioned here is not correct according to Fig. 2. Please check it carefully.

Response: Revised it.

13. Line 261: please add ref. here to show where is this data from

Response: We added it.

14. Line 297: check spell of letters.

Response: We checked and revised them.

[revised manuscript text omitted]
. All the results were plotted including averages and standard deviation values (SD). Additionally, a Pearson correlation and Principal Component Analysis (PCA) was also performed to determine first correlations among the measured variables and reduce the number of studied factors for observing the possible interaction between soil erosion and properties. The raw datasets were standardized before analyses and all statistical analyses were conducted using the R software version 3.2.4 (R Core Team 2013).

**3. Results**

**3.1 Variation of $^{137}$Cs and soil physicochemical properties at different hillslope position**

Our results showed that $^{137}$Cs concentration was significantly different at different hillslope positions (P<0.05) (Fig. 2). The average $^{137}$Cs concentration was the highest in the backslope (0.83±0.54 Bq kg$^{-1}$), followed by the footslope (0.58±0.23 Bq kg$^{-1}$) and shoulder (0.20±0.09 Bq kg$^{-1}$). The $^{137}$Cs inventories at different hillslope positions were respectively 364.6 Bq m$^{-2}$, 249.9 Bq m$^{-2}$and 85.1 Bq m$^{-2}$, and the mean $^{137}$Cs inventories were 226.5 Bq m$^{-2}$.

Similar to the $^{137}$Cs concentration, we found the maximum values of soil pH, SOC, TN, TP, TK in the backslope. Also, it is important to remark that soil pH, TN, TK were higher in the shoulder than in the footslope, meanwhile, SOC and TP were higher in the lower parts than in the upper ones. Except for SOC, other soil properties (pH, TN, TP and TK) were significantly different at different hillslope positions (Table 1).

[Figure]

[Figure]

$^{137}$Cs concentration Bq kg$^{-1}$

**Figure 2:137Cs concentration distribution features at different hillslope positions.**

**Table 1: Average variations in $^{137}$Cs and soil properties at different hillslope positions.**

| | $^{137}$Cs
Bq kg$^{-1}$ | pH | SOM
g kg$^{-1}$ | TN
g kg$^{-1}$ | TP
g kg$^{-1}$ | TK
g kg$^{-1}$ | Slope
gradient (°) |
|---|---|---|---|---|---|---|---|
| Shoulder | 0.20±0.09b | 6.87±0.37b | 0.95±0.72a | 0.09±0.04b | 0.02±0.00b | 1.06±0.21b | 27 |
| Backslope | 0.82±0.54a | 7.47±0.74a | 1.27±0.52a | 0.14±0.05a | 0.03±0.01a | 1.95±0.54a | 20 |
| Footslope | 0.58±0.23a | 6.09±1.03c | 1.05±0.32a | 0.09±0.03b | 0.03±0.01b | 0.95±0.34b | 23 |

Data represent means and standard deviations (SD). Different lowercase letters indicate a significant difference among slope position.

**3.2 Variation of $^{137}$Cs and soil physicochemical properties for three hillslopes positions at different soil depths**

[revised manuscript text omitted]

The Pearson correlation and principal component analysis (PCA) were carried out considering the above-mentioned variables

220    related to sediment deposition rates using [137]Cs. The highest linear correlation was found for SOM (0.669), N (0.643) and P

(0.620). Fig. 4 showed a plot of the eigenvector in the plane of the first two components together with the PC scores in the

plane of PC1 and PC2. On the first component, which explained 61.4% of the total variance, and the second component

explained 23.6%, respectively. Sediment deposition rates (Cs) was significantly affected by hillslope positions, meanwhile,

Cs closely related to SOM and TN. We included the linear correlation between soil erosion rates and SOM, where it can be

225    noted the trend and relationship between both variables (Figure 5).

[Figure]

**Figure 4: Eigenvectors from the principal component analysis (PCA) of the first two components.**

[Figure]

**Figure 5: Linear correlation between $^{137}$Cs and SOM (Soil Organic Matter) content.**

**3.5 Soil erosion modulus estimation**

The reference soil value considered for this estimations were the bulk soil samples without distinguishing any layer. The average $^{137}$Cs concentration in reference site was 6.28 Bq kg$^{-1}$. The surface area of each sample plot was 0.01 m$^2$ and the average weight for the fine particles was 1.5 kg. Using Eq. (2), we obtained 942 Bq m$^{-2}$ as the reference $^{137}$Cs inventory. The $^{137}$Cs

inventory at different hillslope positions (shoulder, backslope, footslope) was lower than the reference plot inventory, possibly indicating that soil erosion happened in the disturbed hillslopes. Using the soil erosion rate Eq. (3) presented above, the calculated soil erosion rates in the shoulder, back- and footslope were 0.87 cm a$^{-1}$, 0.35 cm a$^{-1}$ and 0.49 cm a$^{-1}$, respectively. Using the soil erosion modulus Eq. (6) and combining with the weight of different hillslope positions to calculate it, the average erosion modulus in the whole hillslope was 632 t km$^{-2}$a$^{-1}$.

In the depression bottom, the $^{137}$Cs distribution was much deeper than that of local $^{137}$Cs reference depth and the plough layer. The $^{137}$Cs inventory was much greater than the $^{137}$Cs reference inventory. Based on the depression bottom's $^{137}$Cs peak concentration (165 cm soil layer), the deposition depth since 1963 was 145 cm because the local plough layer is about 20 cm. From Eq. (4) and Eq. (5), we obtained that the average deposition rates since 1963 were 2.68cm a$^{-1}$. Finally, the soil deposition modulus was 3216 t km$^{-2}$a$^{-1}$, calculated from Eq. (6).

**4. Discussion**

Choosing a reference plot was a critical step for assessing the soil erosion rates using $^{137}$Cs fingerprinting technique. While some researchers doubted this method because of $^{137}$Cs fallout heterogeneity (Parsons and Foster, 2011). It is well-known that $^{137}$Cs fallout can be influenced by rainfall and latitude. In our research, the reference site is located only some kilometres away from our plot site and the rainfall and latitude in the reference site are consistent with the plot sites. By calculating, the reference inventory in our study site was 942 Bq m$^{-2}$, close to the values of previous studies under similar rainfall and latitude conditions. For example, Zhang et al. (2009) used 918 Bq m$^{-2}$ as the ideal reference inventory plot in Dianchi watershed Yunnan province. Similarly, Xiong et al (2018) used 906 Bq m$^{-2}$ as the reference inventory in Shilin county, Yunnan province. This period is rather known as the onset of $^{137}$Cs fallout than a high fallout deposition year. We can advertise that a large uncertainty on the reference 1963 fallout peak position (somewhere between 150-200 cm soil depth) could be interpreted. Possibly, it can be due to soil particle's mixing if the land was cultivated or to a more complex deposition trend including a varying supply of $^{137}$Cs-tagged soil particles. Accordingly, any deposition rate that can be derived using this soil depth may be questioned. Therefore, in the future, more soil samples or in situ measures must be performed to clarify this uncertainty.

While soil erosion is usually greatest in the shoulder, followed by the backslope and lowest in the footslope (Song et al., 2018), in our study, soil erosion was greater in either upper and lower hillslope parts than that in the middle one. A steeper gradient of the upper slope compared to the middle could be responsible for this finding. A previous study shows that the slope gradient is the key factor affecting soil erosion when the rainfall and vegetation coverage is remaining unchanged (Lu et al., 2016). On the other hand, other studies considered that steep bare soils subjected to high-intensity rainfall and soil surface features could be rapidly transformed from a loose seedbed to crusted surfaces with pronounced micro-relief, for example, in northern Laos (Ribolzi et al., 2011). In our research area, possibly, the geological or vegetation conditions can also affect in some different ways, therefore, they do not coincide with other published research. ". Possibly, the correlations observed in our study and

reduction of factors using the PCA may not be directly due to $^{137}$Cs adsorption by soil organic matter but rather to the fact that soil micro-aggregates contain both organic matter and $^{137}$Cs bound to fine clay minerals. On the long term a single process, i.e. erosion, would deplete topsoil horizons in both soil organic matter and particle's bound $^{137}$Cs during soil aggregate breakdown. In backslope, chemical dissolution is strong, forming some relatively closed microtopography, such as lapies and solution pans.

270     Soil from the shoulder is easily deposited in these micro topographical forms (Zhang et al., 2009b). Another possible reason is that the coexistence between tillage and water erosion which could play a key role as the main cause of soil loss at the concave position (Lobb and Kachanoski, 1999), i.e. shoulder position (upper parts), while water erosion could lead to serious soil loss at lower hillslope position (these areas used to receive maximum runoff concentrations). 
[revised manuscript text omitted]

485